# Map as a Prompt: Learning Multi-Modal Spatial-Signal Foundation Models for Cross-scenario Wireless Localization

**Yong Chu**[1,2], **Xun Zhou**[1,2,*], **Zenglin Xu**[3,4], **Hui Wang**[2], **Yue Yu**[2]
[1]Harbin Institute of Technology, Shenzhen
[2]Pengcheng Laboratory
[3]Shanghai Academy of AI for Science
[4]Artificial Intelligence Innovation and Incubation Institute, Fudan University
`chuyong@stu.hit.edu.cn, zhouxun2023@hit.edu.cn`

## Abstract

Accurate and robust wireless localization is a critical enabler for emerging 5G/6G applications, including autonomous driving, extended reality, and smart manufacturing. Despite its importance, achieving precise localization across diverse environments remains challenging due to the complex nature of wireless signals and their sensitivity to environmental changes. Existing data-driven approaches often suffer from limited generalization capability, requiring extensive labeled data and struggling to adapt to new scenarios. To address these limitations, we propose SigMap, a multimodal foundation model that introduces two key innovations: (1) A cycle-adaptive masking strategy that dynamically adjusts masking patterns based on channel periodicity characteristics to learn robust wireless representations; (2) A novel "map-as-prompt" framework that integrates 3D geographic information through lightweight soft prompts for effective cross-scenario adaptation. Extensive experiments demonstrate that our model achieves state-of-the-art performance across multiple localization tasks while exhibiting strong zero-shot generalization in unseen environments, significantly outperforming both supervised and self-supervised baselines by considerable margins.

## 1 Introduction

Wireless localization has evolved from classical model-based methods to data-driven deep learning approaches, and more recently, to paradigms built upon foundation models and large language models (LLMs). Despite these advances, existing techniques continue to face significant challenges in complex environments—particularly under non-line-of-sight (NLoS) conditions and in rich multipath scenarios—due to limitations in representation learning and environmental reasoning.

Traditional localization systems rely on geometric or signal-strength measurements such as time-of-arrival (ToA), time-difference-of-arrival (TDoA), angle-of-arrival (AoA), and received signal strength (RSS) (Chen et al., 2022a). Classical algorithms including MUSIC and OMP are widely adopted for parameter estimation (Keskin et al., 2021). However, such model-based methods assume idealized propagation conditions and perform poorly in urban settings with substantial multipath and NLoS effects, often incurring errors over 100 meters (Chen et al., 2024). Although some works attempt to mitigate NLoS via filtering or hardware enhancements (Huang et al., 2023; Zhou et al., 2019), they typically overlook richer environmental semantics from maps or channel characteristics.

To address these issues, data-driven methods have been extensively explored. Supervised models such as MLPs (Gao et al., 2023), CNNs (Wu et al., 2021), and LSTMs (Chen et al., 2023) learn direct mappings from channel state information (CSI) to user positions. While effective in specific settings, they require large labeled datasets and exhibit limited cross-environment generalization (Pan et al., 2025). Subsequent semi-supervised and unsupervised approaches—using autoencoders,

---

GANs, and domain adaptation (Ruan et al., 2023; Chen et al., 2022b; Junoh & Pyun, 2024; Li et al., 2021)—aim to reduce labeling costs, yet often fail to learn robust and transferable representations that capture high-level semantic features of the environment.

Recent efforts have turned toward self-supervised learning (SSL) and foundation models inspired by successes in NLP and vision. Methods such as LWM (Alikhani et al., 2024) and WirelessGPT (Yang et al., 2025) employ masked channel modeling to learn general-purpose channel representations, while contrastive learning frameworks (Salihu et al., 2024) extract invariant channel features. However, these models are not designed specifically for localization and often lack task-aware semantic understanding. Several SSL-based frameworks target localization more directly, including CrowdBERT (Han et al., 2024) and signal-guided masked autoencoders (Wang et al., 2025), which adopt masking strategies for reconstructing RSS or channel impulse responses (CIR). Despite their potential, such approaches are often confined to specific configurations and rely on single SSL objectives, limiting the diversity and generalizability of the learned features.

Concurrently, LLMs have been introduced to the wireless domain. For example, WirelessLLM (Shao et al., 2024) incorporates domain knowledge via prompt engineering and retrieval-augmented generation. Although effective for high-level protocol reasoning, LLMs struggle with low-level signal processing and often produce hallucinations when applied to channel-based inference, restricting their applicability to precise localization tasks.

## 1.1 RESEARCH GAPS

Current wireless localization methods face two fundamental limitations:

- **Inadequate Handling of Signal Periodicity:** Existing self-supervised approaches employ generic masking strategies that ignore the inherent cyclic patterns in Channel State Information (CSI). This allows models to exploit local periodic shortcuts rather than learning meaningful global representations of signal propagation.

- **Superficial Geographic Integration:** While some methods incorporate basic map data, they fail to capture the rich spatial-topological relationships in 3D environments. The fusion between geometric constraints and channel representations remains shallow and lacks interpretability.

## 1.2 CONTRIBUTIONS

This work addresses these gaps through three key contributions:

- **Cycle-Adaptive Masked Modeling:** We introduce a novel masking strategy that dynamically adapts to CSI periodicity by computing row-wise cross-correlation and generating shift-aware patterns. This disrupts periodic shortcuts and forces learning of globally meaningful signal representations.

- **Map-Conditioned Prompt Tuning:** We develop a geographic prompt mechanism that encodes 3D map information via graph neural networks. These prompts enable interpretable fusion of environmental constraints during fine-tuning, enhancing accuracy in complex multipath scenarios.

- **Parameter-Efficient Generalization:** Our foundation model achieves state-of-the-art performance with limited labeled data and demonstrates strong zero-shot generalization to unseen environments and base station configurations.

## 2 PRELIMINARIES

This section introduces the core concepts of wireless channel modeling and formally defines the localization problem. The physical principles explained here are directly leveraged by our geographic prompt tuning method.

## 2.1 WIRELESS CHANNEL MODELING FOR LOCALIZATION

The fundamental premise of our work is that Channel State Information (CSI) contains geometric relationships between transmitters and receivers. As shown in Figure 1, wireless signals propagate through Line-of-Sight (LoS) and Non-Line-of-Sight (NLoS) paths, creating unique spatial fingerprints in the CSI data.

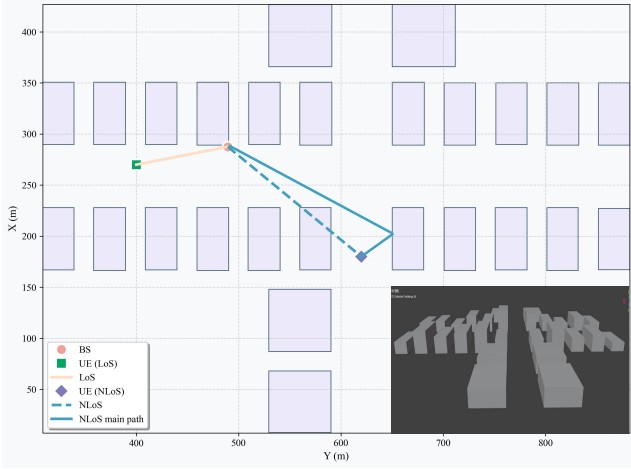

Figure 1: Wireless propagation paths in urban environments. LoS represents direct propagation, while NLoS paths result from reflections and diffractions. **Bottom-right inset**: corresponding 3D map visualization showing the same LoS/NLoS topology used for prompt generation.

For a MIMO-OFDM system with $N_t$ transmit antennas and $N_r$ receive antennas, the CSI matrix $\mathbf{H}[k] \in \mathbb{C}^{N_r \times N_t}$ at subcarrier $k$ can be expressed as the superposition of both LoS and NLoS components:

$$\mathbf{H}[k] = \underbrace{\alpha_{\mathrm{LoS}}e^{-j2\pi\tau_{\mathrm{LoS}}f_k}\mathbf{a}_r(\theta_{\mathrm{LoS}}^r)\mathbf{a}_t(\theta_{\mathrm{LoS}}^t)^H}_{\text{LoS component}} + \underbrace{\sum_{l=1}^{L_{\mathrm{NLoS}}}\alpha_l e^{-j2\pi\tau_l f_k}\mathbf{a}_r(\theta_l^r)\mathbf{a}_t(\theta_l^t)^H}_{\text{NLoS components}} \tag{1}$$

where $L_{\mathrm{NLoS}}$ denotes the number of NLoS multipath components, $\alpha_l$ and $\tau_l$ represent the complex gain and delay of the $l$-th path, $f_k$ is the frequency of the $k$-th subcarrier, and $\mathbf{a}_r(\theta_l^r)$, $\mathbf{a}_t(\theta_l^t)$ are the array steering vectors at receiver and transmitter, respectively.

The key insight for localization is that each path in equation equation 1 carries geometric information. Single-base station localization is possible because CSI contains time delay (related to distance) and angle information that can define a spatial vector. Multipath effects provide multiple such constraints, enabling rough positioning even without precise time measurements. Multi-base station setups provide richer information by offering diverse spatial perspectives.

## 2.2 RAY-TRACING AND MAP ALIGNMENT

We use ray-tracing to generate realistic training data that captures the mapping between physical geometry and wireless channels. The process can be abstracted as:

$$(\mathbf{p}_{\mathrm{BS}}, \mathbf{p}_{\mathrm{UE}}, \mathcal{M}) \xrightarrow{\text{Ray-tracing}} \mathbf{H}_{\mathrm{CSI}} \tag{2}$$

where $\mathbf{p}_{\mathrm{BS}}$ is the base station position, $\mathbf{p}_{\mathrm{UE}}$ is the user position, and $\mathcal{M}$ is the 3D environment map.

The map $\mathcal{M}$ serves two crucial purposes: 1) generating physically realistic training data, and 2) providing geometric constraints during inference to resolve multipath ambiguity. This alignment process helps decompose raw CSI into its constituent LoS and NLoS components, which is learned implicitly by our model through geographic prompt tuning.

### 2.3 PROBLEM FORMULATION: WIRELESS LOCALIZATION

We define the wireless localization problem as estimating user equipment position from channel measurements and environmental context.

**Inputs**:

- Channel State Information $\mathbf{H} \in \mathbb{C}^{N_r \times N_t \times N_{sc}}$ from one or multiple base stations
- 3D environment map $\mathcal{M}$ containing building geometries
- Base station positions $\mathbf{P}_{\text{BS}} = \{\mathbf{p}_{\text{BS}}^{(1)}, \ldots, \mathbf{p}_{\text{BS}}^{(T)}\}$

**Output**: Estimated user position $\hat{\mathbf{p}}_{\text{UE}} \in \mathbb{R}^3$.

**Objective**: Learn a mapping function $f_\theta$ that minimizes:

$$\mathbb{E}\left[\|f_\theta(\mathbf{H}, \mathcal{M}, \mathbf{P}_{\text{BS}}) - \mathbf{p}_{\text{UE}}\|^2\right] \tag{3}$$

The inclusion of map information $\mathcal{M}$ differentiates our approach from conventional CSI-only methods, enabling more accurate and physically consistent localization.

## 3 METHODOLOGY

### 3.1 OVERALL FRAMEWORK

Our proposed wireless localization foundation model addresses the fundamental challenge of achieving accurate positioning across diverse environments with minimal labeled data requirements. The framework follows a two-stage learning paradigm consisting of self-supervised pre-training on unlabeled CSI data followed by prompt-based fine-tuning for specific localization tasks. This approach enables the model to learn general-purpose representations of wireless signal propagation that can be efficiently adapted to new environments.

As illustrated in Figure 2, the framework integrates three core components: (1) a transformer-based backbone network that captures long-range dependencies in CSI data, (2) a novel cycle-adaptive masked modeling strategy that prevents shortcut learning in periodic signals, and (3) a geographic prompt tuning mechanism that incorporates environmental constraints during fine-tuning. The key innovation lies in our cycle-aware masking approach that dynamically adapts to signal periodicity, combined with map-conditioned prompts that enable efficient adaptation with minimal parameter updates.

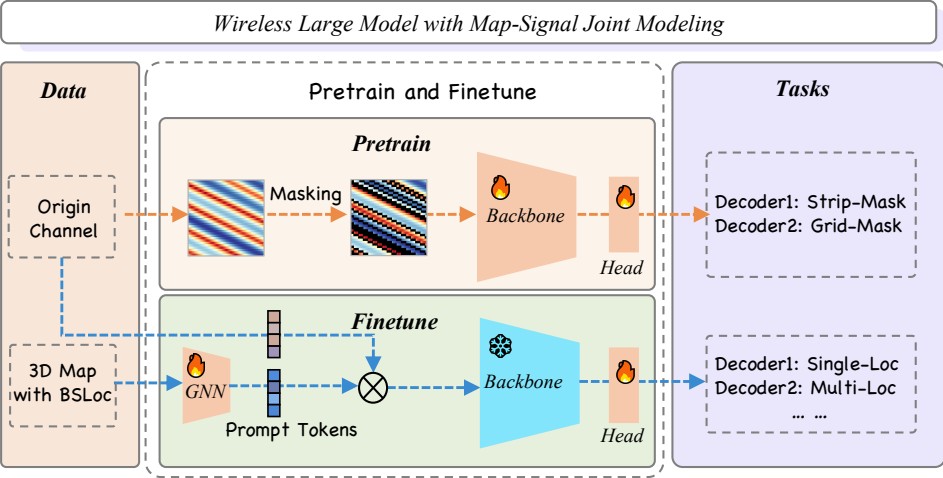

Figure 2: Overall architecture of our wireless localization foundation model, showing the two-stage learning process with self-supervised pre-training and prompt-based fine-tuning.

## 3.2 SIGNAL REPRESENTATION AND PREPROCESSING

Wireless Channel State Information (CSI) provides a rich characterization of the propagation environment by capturing multipath effects, fading characteristics, and spatial diversity. In multi-antenna OFDM systems, we represent the channel frequency response as a complex-valued tensor:

$$\mathcal{H} \in \mathbb{C}^{N_r \times N_t \times N_s} \tag{4}$$

where $N_r$, $N_t$, and $N_s$ denote the number of receive antennas, transmit antennas, and subcarriers respectively. Each element $h_{i,j}[k]$ represents the complex channel gain between specific antenna pairs at different subcarriers.

To facilitate deep learning processing while preserving critical phase information, we transform the complex CSI data into a real-valued representation through channel-wise separation:

$$\mathbf{X} = [\Re(\mathcal{H}), \Im(\mathcal{H})] \in \mathbb{R}^{2 \times N_r \times N_t \times N_s} \tag{5}$$

This representation maintains the spatial and frequency diversity essential for accurate localization while being compatible with standard neural network operations.

## 3.3 CYCLE-ADAPTIVE MASKED MODELING

Traditional masked autoencoding approaches often struggle with wireless signals due to their inherent periodic patterns, which can be exploited as learning shortcuts. Our cycle-adaptive masking strategy addresses this limitation by dynamically generating mask patterns that disrupt periodic structures while preserving semantically meaningful information.

The core insight is to detect dominant periodicities in the CSI data and generate masks that prevent simple interpolation-based reconstruction. For each input sample, we compute shift patterns using cross-correlation analysis and generate adaptive mask patterns:

$$\mathbf{M}_{\text{cycle}}[i,j] = \begin{cases} 0 & \text{if } |j - (j_0 + i \cdot d_{\text{final}})| \leq w \\ 1 & \text{otherwise} \end{cases} \tag{6}$$

where $d_{\text{final}}$ represents the detected periodicity shift, $j_0$ is the starting offset, and $w$ controls the mask width. This approach ensures that the model must learn meaningful signal representations rather than relying on pattern repetition, as illustrated in Figure 3.

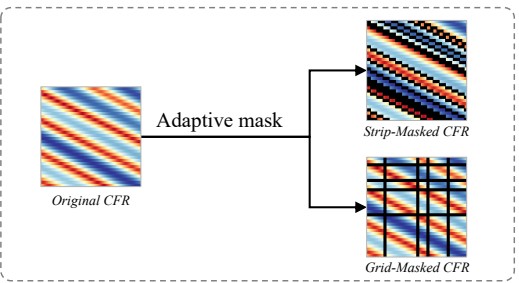

Figure 3: Illustration of our cycle-adaptive masking strategy. The mask pattern (right) is dynamically generated based on the detected periodicity in the CSI amplitude data (left), preventing the model from exploiting simplistic periodic shortcuts.

The reconstruction objective trains the model to recover the original signal from masked inputs:

$$\mathcal{L}_{\text{MAE}} = \mathbb{E}_{\mathbf{X}} \left[ \| \mathbf{X} - f_{\theta_{\text{dec}}}(\mathbf{X}_{\text{masked}}) \|^2 \right] \tag{7}$$

### 3.4 GEOGRAPHIC PROMPT TUNING

Following pre-training, we employ a parameter-efficient fine-tuning strategy that leverages geographic information from 3D environment models. The core innovation is the transformation of spatial relationships between buildings and base stations into a set of learnable prompt tokens that guide the pre-trained model without updating its core parameters.

---

**Algorithm 1** Geographic Prompt Generation

---

1: **procedure** GENERATEGEOPROMPT($\mathcal{M}, \mathbf{P}_{\text{BS}}$)
2:      $\mathbf{h}_v^{(0)} = \text{MLP}_{\text{vert}}(\mathbf{v}; \mathbf{W}_{\text{vert}})$              ▷ Encode vertex positions
3:      $\mathbf{h}_{\text{BS}}^{(0)} = \text{MLP}_{\text{BS}}(\mathbf{p}_{\text{BS}}; \mathbf{W}_{\text{bs}})$             ▷ Encode BS positions
4:      $\mathcal{V}_{\text{init}} = \{\mathbf{h}_v^{(0)}\} \cup \{\mathbf{h}_{\text{BS}}^{(0)}\}$
5:      **for** $l = 1$ to $2$ **do**                   ▷ 2 Graph convolution layers
6:          **for** $i \in \mathcal{V}$ **do**
7:              $\mathbf{h}_i^{(l)} = \sigma\left(\mathbf{W}^{(l)}\mathbf{h}_i^{(l-1)} + \sum_{j \in \mathcal{N}(i)} \mathbf{U}^{(l)}\mathbf{h}_j^{(l-1)}\right)$
8:          **end for**
9:      **end for**
10:     $\mathbf{g} = \text{GlobalMeanPool}(\{\mathbf{h}_i^{(2)}\}_{i=1}^{|\mathcal{V}|})$        ▷ Aggregate graph information
11:     $\mathbf{g}_{\text{prompt}} = \text{MLP}_{\text{proj}}(\mathbf{g}; \mathbf{W}_{\text{proj}})$          ▷ Project to prompt dimension
12:     **return** $\mathbf{g}_{\text{prompt}}$
13: **end procedure**

---

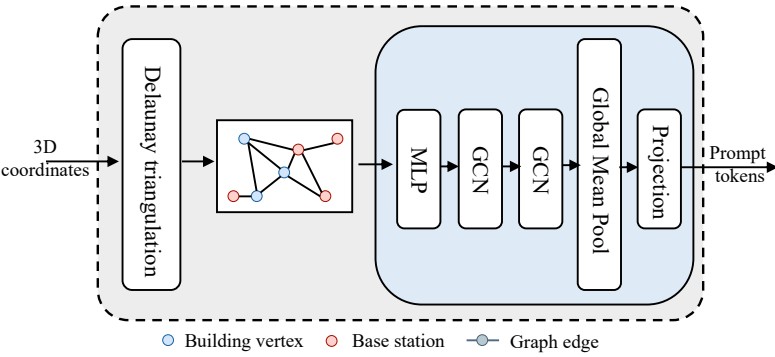

Figure 4: The pipeline of geographic prompt generation.

The process begins with the construction of a heterogeneous graph $\mathcal{G} = (\mathcal{V}, \mathcal{E})$ that encodes the spatial configuration of a given scene. The scene is defined by a 3D building mesh $\mathcal{M}$, represented by a set of vertices $\{\mathbf{v}_i\}_{i=1}^V$ where each $\mathbf{v}_i \in \mathbb{R}^3$, and the positions of $T$ base stations, denoted as $\mathbf{P}_{\text{BS}} = \{\mathbf{p}^t\}_{t=1}^T$ where each $\mathbf{p}^t \in \mathbb{R}^3$. The node set of the graph is the union of these building vertices and base station positions: $\mathcal{V} = \{\mathbf{v}_1, \ldots, \mathbf{v}_V\} \cup \{\mathbf{p}^1, \ldots, \mathbf{p}^T\}$. To capture the inherent proximity relationships in 3D space, the edge set $\mathcal{E}$ is constructed using Delaunay triangulation over the node set $\mathcal{V}$, formally defined as $\mathcal{E} = \{(i, j) \mid \text{nodes } i \text{ and } j \text{ are connected in the Delaunay triangulation of } \mathcal{V}\}$.

The GCN update rule for each layer is formally defined as:

$$\mathbf{H}^{(l+1)} = \sigma\left(\tilde{\mathbf{D}}^{-\frac{1}{2}}\tilde{\mathbf{A}}\tilde{\mathbf{D}}^{-\frac{1}{2}}\mathbf{H}^{(l)}\mathbf{W}^{(l)}\right)$$

where $\tilde{\mathbf{A}} = \mathbf{A} + \mathbf{I}$ is the adjacency matrix with self-connections, $\tilde{\mathbf{D}}$ is the degree matrix of $\tilde{\mathbf{A}}$, and $\mathbf{W}^{(l)}$ are the trainable weights of layer $l$.

The generated geographic prompt $\mathbf{g}_{\text{prompt}} \in \mathbb{R}^{D_p}$ is integrated into the pre-trained Transformer's input sequence. The complete input sequence $\mathbf{T}_{\text{input}} \in \mathbb{R}^{(1+1+L)\times D}$ is constructed by prepending the prompt to the existing sequence. It consists of the frozen classification token $\mathbf{t}_{\text{cls}}$, the trainable geographic prompt token $\mathbf{T}_{\text{geo}} = \mathbf{g}_{\text{prompt}}$ (for a single prompt), and the frozen sequence of CSI

measurement tokens $\mathbf{T}_{\text{CSI}}$. This combined sequence is then added to the frozen positional encoding $\mathbf{E}_{\text{pos}}$:

$$\mathbf{T}_{\text{input}} = [\mathbf{t}_{\text{cls}}; \mathbf{T}_{\text{geo}}; \mathbf{T}_{\text{CSI}}] + \mathbf{E}_{\text{pos}}$$

The self-attention mechanism then operates on this extended sequence. The Query ($\mathbf{Q}$), Key ($\mathbf{K}$), and Value ($\mathbf{V}$) matrices are derived by projecting the input sequence with the frozen pre-trained weight matrices $\mathbf{W}^Q$, $\mathbf{W}^K$, and $\mathbf{W}^V$:

$$\mathbf{Q} = \mathbf{T}_{\text{input}}\mathbf{W}^Q, \quad \mathbf{K} = \mathbf{T}_{\text{input}}\mathbf{W}^K, \quad \mathbf{V} = \mathbf{T}_{\text{input}}\mathbf{W}^V$$

The attention output is computed as $\text{Attention}(\mathbf{Q}, \mathbf{K}, \mathbf{V}) = \text{softmax}\left(\frac{\mathbf{Q}\mathbf{K}^T}{\sqrt{d_k}}\right)\mathbf{V}$.

The parameter efficiency of this approach is a key advantage. The only parameters updated during fine-tuning are those of the GNN ($\theta_{\text{gnn}}$), the projection MLP ($\theta_{\text{proj}}$), and the task-specific head ($\theta_{\text{task}}$). The optimization process is formulated as:

$$\min_{\theta_{\text{gnn}},\theta_{\text{proj}},\theta_{\text{task}}} \mathbb{E}_{(\mathbf{X},\mathcal{M},\mathbf{P}_{\text{BS}},\mathbf{y})\sim\mathcal{D}_{\text{task}}} \left[\mathcal{L}_{\text{task}}(f(\mathbf{X},\mathcal{M},\mathbf{P}_{\text{BS}}),\mathbf{y})\right]$$

where the complete forward pass is defined as $f(\mathbf{X}, \mathcal{M}, \mathbf{P}_{\text{BS}}) = f_{\theta_{\text{task}}}(f_{\theta_{\text{enc}}}([\mathbf{T}_{\text{geo}}; \mathbf{T}_{\text{CSI}}]))$.

### 3.5 TASK-SPECIFIC ADAPTATION

We design specialized output heads to handle different localization scenarios. For single-base station localization, the user equipment position is directly predicted from the final [CLS] token using a simple MLP head:

$$\hat{\mathbf{p}}_{\text{UE}} = \text{MLP}_{\text{single}}(\mathbf{t}_{\text{cls}}; \mathbf{W}_{\text{single}}) \tag{8}$$

For multi-base station scenarios, we employ an attention-based fusion mechanism that dynamically integrates information from all available base stations. The process begins by extracting the [CLS] tokens from all $T$ base stations and stacking them into a tensor $\mathbf{B} \in \mathbb{R}^{T \times D}$. We then compute attention weights $\alpha_t$ for each base station using a learned attention function:

$$\alpha_t = \frac{\exp(\mathbf{v}^T \tanh(\mathbf{W}_{\text{attn}}\mathbf{t}_{\text{cls}}^{(t)}))}{\sum_{j=1}^{T} \exp(\mathbf{v}^T \tanh(\mathbf{W}_{\text{attn}}\mathbf{t}_{\text{cls}}^{(j)}))} \tag{9}$$

Each base station's [CLS] token is processed independently through dedicated MLP heads to generate preliminary position estimates $\hat{\mathbf{p}}_{\text{UE}}^{(t)}$. The final position estimate is obtained through weighted fusion:

$$\hat{\mathbf{p}}_{\text{UE}} = \sum_{t=1}^{T} \alpha_t \cdot \text{MLP}_{\text{multi}}^{(t)}(\mathbf{t}_{\text{cls}}^{(t)}; \mathbf{W}_{\text{multi}}^{(t)}) \tag{10}$$

This attention mechanism allows the model to dynamically prioritize contributions from different base stations based on their signal quality and geometric configuration, with stations having stronger signals or more favorable geometric relationships receiving higher weights. The comprehensive framework demonstrates how self-supervised pre-training combined with geographic-aware prompt tuning can achieve robust wireless localization across diverse environments while maintaining parameter efficiency and practical deployability.

## 4 EXPERIMENTS

We conduct comprehensive experiments to evaluate our wireless localization foundation model across diverse scenarios. The experiments address four key questions: (1) How does our method compare to state-of-the-art approaches? (2) What is the impact of geographic information? (3) How effective is our cycle-adaptive masking? (4) How well does our method generalize to new environments?

## 4.1 DATASETS AND EVALUATION METRICS

We evaluate our method on the DeepMIMO dataset (Alkhateeb, 2019), using the O1_3p5 urban scenario for both pre-training and fine-tuning. The dataset provides realistic CSI data generated through ray-tracing simulations. Detailed configuration parameters are provided in Appendix B.3.

Evaluation metrics include Mean Absolute Error (MAE), Root Mean Square Error (RMSE), and Cumulative Distribution Function at 1 meter (CDF@1m). All results are averaged over 5 independent runs.

## 4.2 MAIN RESULTS

We compare against OMP (compressed sensing), CNN-based, SWiT (Salihu et al., 2024), and LWLM (Pan et al., 2025).

Single-BS localization under NLoS represents one of the most challenging scenarios. As shown in Table 1, SIGMAP with geographic information achieves an MAE of $1.564\,\mathrm{m}$, RMSE of $5.675\,\mathrm{m}$, and CDF@1m of $60.5\%$, outperforming the best baseline (LWLM) by $34.4\%$ in MAE and more than doubling the CDF@1m.

The key advantage stems from our NLoS-aware attention mechanism that explicitly models multipath propagation:

$$\alpha_i = \frac{\exp\big(\phi\big(\boldsymbol{o}_s^{(i)} \cdot \mathbf{W}_{\mathrm{NLoS}}\big)\big)}{\sum_j \exp\big(\phi\big(\boldsymbol{o}_s^{(j)} \cdot \mathbf{W}_{\mathrm{NLoS}}\big)\big)}, \tag{11}$$

which allows the model to differentiate between direct and reflected paths, significantly reducing positioning ambiguity.

Table 1: Metrics of Single-BS localization.

| Method | MAE (m) | RMSE (m) | CDF@1m (%) |
|---|---|---|---|
| SIGMAP (w/ map) | **1.564** | **5.675** | **60.5** |
| SIGMAP (w/o map) | 2.275 | 8.532 | 31.0 |
| LWLM | 2.382 | 5.822 | 25.3 |
| SWiT | 2.586 | 8.967 | 24.3 |
| CNN | 2.943 | 9.423 | 21.7 |
| OMP | 3.287 | 9.851 | 15.4 |

Multi-BS collaboration leverages spatial diversity to overcome NLoS limitations. Table 2 shows that SIGMAP with map achieves $0.673\,\mathrm{m}$ MAE, $1.099\,\mathrm{m}$ RMSE and $84.5\%$ CDF@1m, improving the second-best result (SIGMAP w/o map) by $14.7\%$ in MAE and 7.0 percentage-points in CDF@1m. As further visualized in Figure 5, SIGMAP dominates accuracy, robustness and precision simultaneously. The CDF curves are shown in B.5

Table 2: Metrics of Multi-BS (4-BS) collaborative localization.

| Method | MAE (m) | RMSE (m) | CDF@1m (%) |
|---|---|---|---|
| SIGMAP (w/ map) | **0.673** | **1.099** | **84.5** |
| SIGMAP (w/o map) | 0.789 | 1.285 | 77.5 |
| LWLM | 0.828 | 1.178 | 75.6 |
| SWiT | 1.102 | 1.368 | 68.1 |
| CNN | 1.398 | 1.731 | 59.3 |
| OMP | 1.685 | 2.089 | 50.6 |

## 4.3 EFFECTIVENESS OF CYCLE-ADAPTIVE MASKING

Table 3 compares masking strategies. Cycle-adaptive masking (last row) yields the best trade-off: $0.673\,\mathrm{m}$ MAE and $84.5\%$ CDF@1m, outperforming fixed grid or strip masking. Dynamic disruption of periodic CSI patterns forces the model to learn generalizable features instead of shortcut interpolation.

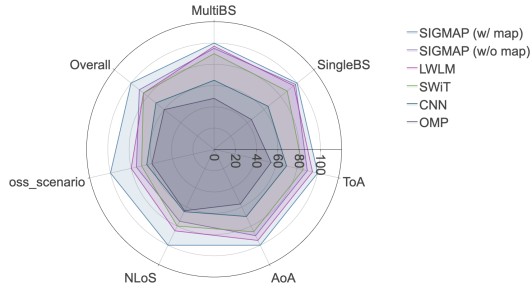

Figure 5: Comprehensive performance comparison across metrics. Our method shows consistent superiority in accuracy and robustness.

Table 3: Effect of cycle-adaptive masking strategy.

| Method | MAE (m) | RMSE (m) | CDF@1m (%) |
|---|---|---|---|
| Grid-masking only | 0.770 | 1.176 | 80.3 |
| Strip-masking only | 0.753 | **0.972** | 75.3 |
| Adaptive masking | **0.673** | 1.099 | **84.5** |

## 4.4 ABLATION STUDY ON MAP PROMPTS

To quantify the influence of map quality on localization accuracy, we conducted an ablative comparison using (i) complete 3-D mesh, (ii) 2-D bird's-eye polygon, and (iii) no-map (CSI-only). Except for height, the 2-D variant follows the same pipeline as the 3-D version; the gap arises solely from missing height and facade normals.

Two-dimensional and three-dimensional map ablations are illustrated side-by-side in Figure 1. The near-overlapping error bars indicate that most of the topological benefit is retained even without vertical detail. This outcome suggests an immediate upgrade path: replacing the 2-D polygon with a street-level photograph (visual prompt) could re-introduce facade and texture cues, offering a low-cost yet effective extension for future work.

Results are summarised in Table 4: the 2-D bird's-eye view degrades MAE by 8 % relative to the full 3-D mesh, confirming that most gain comes from topological/LoS cues and that the prompt mechanism is robust to moderate geometric simplification.

Table 4: Single-BS localization with different map modalities.

| Method | MAE (m) | RMSE (m) | CDF@1m (%) |
|---|---|---|---|
| SIGMAP (3-D map) | **1.564** | **5.675** | **60.5** |
| SIGMAP (2-D birdview) | 1.692 | 6.128 | 55.7 |
| SIGMAP (w/o map) | 2.275 | 8.532 | 31.0 |

## 4.5 GENERALIZATION TO NEW ENVIRONMENTS

We evaluate generalization on two completely unseen ray-tracing suites: (i) DeepMIMO_O2 scenario and (ii) WAIR-D (Huangfu et al., 2022) Scenario-2 (100 real-world city scenes extracted from OpenStreetMap). Typical complex scenes of WAIR-D are shown in Figure 6, illustrating dense urban canyons and irregular footprints that challenge cross-domain transfer.

In all experiments, only the downstream task heads are fine-tuned using limited target samples (approximately 100 instances per scenario), while the self-supervised backbone remains frozen. This few-shot learning setup demonstrates the method's ability to rapidly adapt to new environments. Results are listed in Table 4.5.

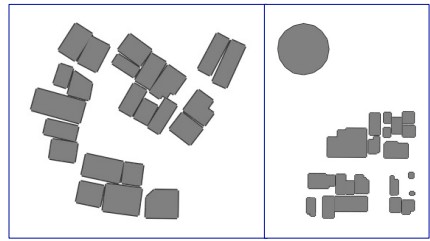

Figure 6: Two typical complex scenes of WAIR-D.

Generalization performance on unseen scenarios with minimal fine-tuning.

| Method | MAE (m) | RMSE (m) | CDF@1m (%) |
|---|---|---|---|
| *DeepMIMO O2 outdoor* | | | |
| SIGMAP (w/ map) | **1.026** | **1.551** | **66.4** |
| SIGMAP (w/o map) | 1.282 | 5.824 | 63.9 |
| LWLM | 2.213 | 11.837 | 63.2 |
| *WAIR-D Scenario-2 (100 cities)* | | | |
| SIGMAP (w/ map) | **1.880** | **3.717** | **58.0** |
| SIGMAP (w/o map) | 2.578 | 4.650 | 51.5 |
| LWLM | 3.375 | 6.921 | 50.3 |

Equipped with geographic prompts, SIGMAP reaches 1.026 m MAE on DeepMIMO O2 and 1.580 m on WAIR-D Scenario-2, outperforming LWLM by 53.2 % and 44.3 %, respectively, while updating only 0.4 % of parameters. The results confirm that, by generating environment-specific prompts, SIGMAP delivers good transfer performance across entirely different wireless environments.

## 4.6 PARAMETER EFFICIENCY

Table 5: Training cost comparison under our experimental setup.

| Stage | Trainable Params | Time/Epoch | Total Time |
|---|---|---|---|
| Pre-train | 11.730 M | 10.8 min | 36 h |
| Fine-tune | 0.085 M | 1.8 s | 30 min |
| Inference | — | 0.83 ms/sample | — |

Under the experimental setup detailed in Appendix B, the model is first pre-trained for 200 epochs and then fine-tuned for 1000 epochs; because only 0.7% of the total parameters are activated during fine-tuning, the entire 1000-epoch fine-tuning stage takes merely 30 min, while still preserving the rich representations learned during pre-training, demonstrating significant parameter efficiency.

## 5 CONCLUSION

This paper presents a wireless localization foundation model that achieves state-of-the-art performance through cycle-adaptive masking and geographic prompt tuning. Our approach delivers strong and consistent accuracy in both single-BS and multi-BS tasks, and generalizes robustly across previously unseen geographic scenarios.

Future work will explore two key directions: extending beyond localization to develop general-purpose wireless foundation models for channel estimation, beamforming and signal processing tasks; and integrating visual modalities such as images and point clouds with wireless signals to create richer environmental representations when 3D maps are incomplete or unavailable.

These advances will lead to more versatile and practical wireless perception systems for emerging applications in smart infrastructure and mobile computing.

ACKNOWLEDGMENTS

This work was partially supported by the National Natural Science Foundation of China (62472125), Guangdong Basic and Applied Basic Research Foundation (2025A1515011258), Key Technologies R&D Program of Guangdong Province (2026B0909060001) and Shenzhen Science and Technology Programs (GXWD20231128102922001, ZDCY20250901111705007, ZDSYS20230626091203008). This work is supported by the Major Key Project of PCL (Grant No. PCL2024A08).

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

## A  STATEMENTS

### A.1  USE OF LLMS STATEMENT

We employed large-language-model tools primarily for language polishing and phrasing suggestions. All technical content, experimental designs, and scientific interpretations were conceived, reviewed, and approved by the authors.

### A.2  REPRODUCIBILITY STATEMENT

To facilitate reproducibility, we have consolidated the complete pipeline—dataset generation, model pre-training, fine-tuning, and evaluation scripts—in an anonymous GitHub repository (`https://anonymous.4open.science/r/SigMap_anonymous-838D`)

## B  DETAILED SETUPS OF OUR EXPERIMENTS

### B.1  COMPUTE RESOURCES

Our experiments were conducted on a computing server equipped with the following specifications:

- **GPUs**: 6 × NVIDIA A800 80GB PCIe
- **GPU Memory**: 80 GB per GPU (480 GB total)
- **Driver Version**: 550.144.03
- **CUDA Version**: 12.4

The A800 GPUs provided the necessary computational power for training large-scale transformer models and processing high-dimensional CSI data with complex shift pattern augmentations.

## B.2 GENERAL CONFIGURATIONS

The input to the model is the complex CFR matrix $\boldsymbol{H}_s$. To facilitate neural network processing, we decompose it into its magnitude and phase components, denoted as $\overline{\boldsymbol{H}}_s$, and rewrite it as

$$\overline{\boldsymbol{H}}_s = [|\boldsymbol{H}_s|, \angle \boldsymbol{H}_s] \in \mathbb{R}^{2 \times N_{\text{ant}} \times N_{\text{subc}}}. \tag{12}$$

The input $\overline{\boldsymbol{H}}_s$ is a 3D tensor of shape $(2, N_{\text{ant}}, N_{\text{subc}})$, with the first dimension corresponding to the amplitude and phase, respectively. Although $\overline{\boldsymbol{H}}_s$ represents a specific format of the channel input, for notational consistency throughout the paper, we will still use $\boldsymbol{H}_s$ to refer to the general representation of the input channel data in all subsequent discussions.

We employed a transformer-based encoder-decoder framework specifically designed for wireless channel modeling and localization tasks. The key architectural components include:

- **Input Dimensions**: $B \times C \times T \times F$ where:
  - $B$: Batch size (32)
  - $C$: Channel dimensions (2 for real/imaginary components)
  - $T$: Time/Antenna dimension (128)
  - $F$: Frequency/Subcarrier dimension (32)
- **Encoder**: Multi-head self-attention layers with positional encoding
- **Decoder**: Cross-attention mechanisms for coordinate prediction
- **Feature Dimension**: 512-dimensional latent representations

Table 6: Training Hyperparameters

| Parameter | Value |
|---|---|
| Batch Size | 32 |
| Optimizer | Adam |
| Learning Rate | $1 \times 10^{-4}$ |
| Weight Decay | $1 \times 10^{-5}$ |
| Training Epochs | 300 |
| Gradient Clipping | 1.0 |
| Learning Rate Schedule | Cosine Annealing |
| Warm-up Epochs | 10 |

## B.3 DATASET PARAMETERS

Table 7: Detailed DeepMIMO dataset configuration parameters

| Parameter | Pre-training | Fine-tuning |
|---|---|---|
| Scenario | O1_3p5 | O1_3p5 |
| Number of BSs | 4 | 4 |
| BS IDs | [3, 4, 9, 10] | [3, 4, 9, 10] |
| Frequency bands (MHz) | [10, 20, 50] | 10 |
| Bandwidth (GHz) | [0.01, 0.02, 0.05] | 0.01 |
| Subcarriers | 128 | 128 |
| Antenna elements | 32 | 32 |
| User distribution | Uniform | Random |
| User subsampling | 100% | 2% |
| Number of paths | 5 | 5 |
| CSI samples | 480,000 | 12,000 |
| Train/Val/Test split | - | 10,000/1,000/10,00 |

The pre-training data was generated using the following key parameters:

- Scenario: O1_3p5 (urban outdoor environment)

- Active base stations: 3, 4, 9, 10
- Frequency bands: 10MHz, 20MHz, 50MHz
- Antenna configuration: 32-element uniform linear array
- Subcarrier configuration: 128 subcarriers, all selected
- User coverage: Complete row coverage (5200 users)

Each sample contains complex channel data (real and imaginary components), user and base station locations, line-of-sight status, distance information, and angle-of-departure parameters.

The fine-tuning dataset shares the same environmental scenario but with different sampling strategy:

- Single frequency band: 10MHz
- User subsampling: 2% of available users
- Data split: 10,000 training, 1,000 validation, 1,000 test samples
- Quality filtering: Only samples with valid path information are included

The dataset ensures comprehensive coverage of the environment while maintaining realistic user distribution patterns for effective model evaluation.

### B.4 DETAILS OF DATA AUGMENTATION EXPERIMENTS

CSI amplitude data often exhibits periodicity due to hardware properties of RF chains, such as antenna spacing and carrier frequency. For instance, in wireless systems with uniform linear arrays, channel responses between antennas may repeat periodically after a fixed number of antennas.

In the process of reconstructing masked channel data using a Vision Transformer (ViT)-based Masked Autoencoder (MAE), masking only individual rows or columns could allow the model to easily learn superficial periodic patterns, thereby failing to capture global features from redundant information.

To address this, we adopted a classical time-series method: computing the cross-correlation coefficient between each row of the Channel Frequency Response (CFR) matrix and the next row. Given that each row has the same length, significant boundary effects emerge. To mitigate this, we restricted comparisons only to valid regions, avoiding boundary artifacts. In the example provided, row shifts vary (e.g., d = 8, d = -3, d = 0, see Fig. 7), which can be positive, negative, or zero. This motivated our adaptive masking strategy.

When the bandwidth equals the row-wise shift amount, adjacent masked bands connect end-to-end, forming visually continuous diagonal strips without gaps. This represents the minimum critical width required to achieve solid and continuous masking. The shift pattern augmentation technique is mathematically formulated as follows:

$$M_{\text{shift}} = \text{GenerateShiftMask}(d, N_a, N_s, T, F) \tag{13}$$

where:

- $d$: Slope parameter controlling shift direction and magnitude
- $N_a$: Number of antenna-based masks (8)
- $N_s$: Number of subcarrier-based masks (32)
- $T$: Time dimension size (128)
- $F$: Frequency dimension size (32)

The shift pattern generation algorithm proceeds through these steps:

1. **Parameter Initialization**:

$$\text{Bandwidth } bw = |d|$$
$$\text{Half-bandwidth } hw = \lfloor bw/2 \rfloor$$
$$\text{Padding } P = hw$$

2. **Matrix Padding**: Expand frequency dimension to accommodate shifts:

$$F_{\text{padded}} = F + 2P \tag{14}$$

3. **Antenna-based Mask Generation**: For $i = 1$ to $N_a$:

$$\text{Start column } c_0 \sim \mathcal{U}(0, F) \tag{15}$$
$$\text{Column positions } c(t) = c_0 - t \cdot d + P \tag{16}$$
$$\text{Mask band } B(t) = [c(t) - hw, c(t) + hw] \cap [0, F_{\text{padded}}] \tag{17}$$

4. **Subcarrier-based Mask Generation**: For $j = 1$ to $N_s$:

$$\text{Start row } r_0 \sim \mathcal{U}(0, T) \tag{18}$$
$$\text{Column positions } c(t) = (t - r_0) \cdot (-d) + P \tag{19}$$
$$\text{Mask band } B(t) = [c(t) - hw, c(t) + hw] \cap [0, F_{\text{padded}}] \tag{20}$$

5. **Mask Application**: The final augmented input is computed as:

$$X_{\text{augmented}} = X \odot M + (1 - M) \odot T_{\text{mask}} \tag{21}$$

where $T_{\text{mask}}$ represents learnable mask tokens.

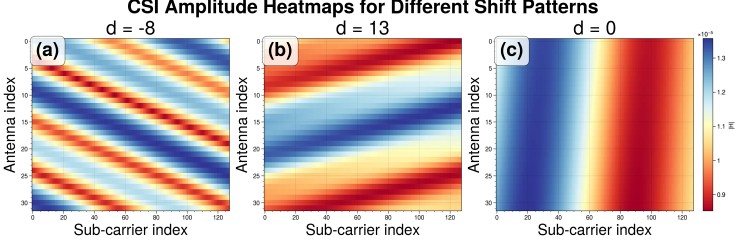

Figure 7: CSI Amplitude Heatmaps for Different Shift Patterns

### B.5 LOCALIZATION ERROR CDF CURVES

The Cumulative Distribution Function (CDF) of localization error measures the probability that the positioning error is less than or equal to a given distance. It is the key metric used in Section 4.2 (Main Results) to compare accuracy and robustness across methods. Figures 8 and 9 plot these CDFs for single-BS and 4-BS collaborative scenarios, respectively. A steeper curve and higher value at 1 m indicate better performance; SIGMAP (w/ map) reaches 60.5% and 84.5% CDF@1m in the two settings, clearly outperforming all baselines.

## C CHARACTERISTICS OF CHANNEL DATA

Wireless Channel State Information (CSI) data exhibits unique characteristics that distinguish it from conventional vision or language modalities and even from generic time series and spatio-temporal data. These traits motivated the antenna–subcarrier joint masking used in SigMap.

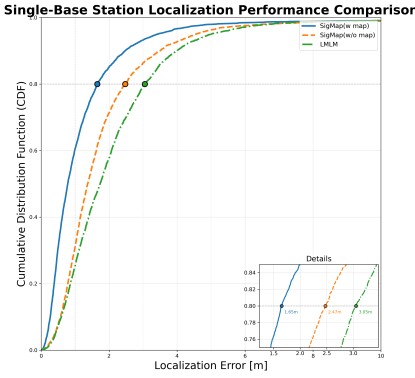

Figure 8: Single-Base Station Localization Performance Comparison

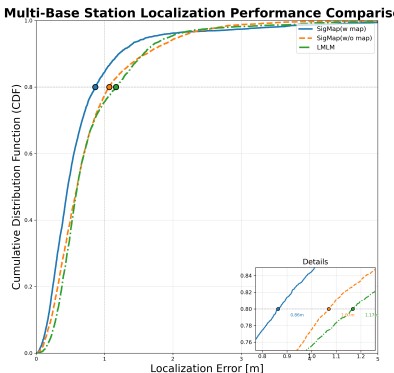

Figure 9: Multi-Base Station Localization Performance Comparison

## C.1 UNIQUE DIMENSIONALITY: SPATIAL-TEMPORAL-SPECTRAL STRUCTURE

A CSI tensor $\mathcal{H}$ captured by a multi-antenna OFDM system is inherently multi-dimensional, spanning three critical domains:

- **Spectral Domain (Subcarriers)**: Represents the frequency-selective fading of the channel. The correlation across subcarriers $k$ is a function of the delay spread $\tau_{\max}$ of the multipath environment, often modeled by the channel's frequency correlation function.

- **Spatial Domain (Antennas)**: Captures the geometric aspects of the propagation. The correlation across antenna elements $n$ is a function of the angle spread $\theta_{\text{spread}}$ and array geometry, described by the spatial correlation matrix $\mathbf{R}_{\text{spatial}} = \mathbb{E}[\mathbf{H}_f \mathbf{H}_f^H]$.

- **Temporal Domain (Snapshots)**: Represents the time-varying nature of the channel due to mobility or environmental changes, characterized by the Doppler spread $f_d$.

This structure can be formalized as a 3D tensor $\mathcal{H} \in \mathbb{C}^{N_{\text{ant}} \times N_{\text{sc}} \times N_{\text{time}}}$, making it a *Spatial-Temporal-Spectral* data cube. This is distinct from:

- **Time Series**: Which are typically 1D ($N_{\text{time}}$) and lack explicit spatial and spectral structure.

- **Spatio-Temporal Data (e.g., traffic grids, videos)**: Which are often 2D+Time ($Height \times Width \times Time$) with spatial homogeneity. The spatial dimensions in CSI are non-grid-like (antenna array geometry) and coupled with the spectral domain.

## C.2 IMPLICATIONS FOR FOUNDATION MODEL DESIGN

The aforementioned characteristics necessitate specialized adaptations in foundation model architecture and pre-training strategies, moving beyond direct applications of models designed for other modalities.

- **Beyond Standard ViT Patches**: While Vision Transformers (ViTs) process images by splitting them into regular 2D patches, this is suboptimal for CSI. Our *cycle-adaptive masking* strategy (Sec. 3.2) is a direct response to this, designed to respect the inherent periodicity and structure within the spatial-spectral planes of the CSI tensor, rather than treating it as a generic image.

- **Beyond Standard MAE for Images**: Masked Autoencoding (MAE) for images relies on the intuition that adjacent pixels are highly correlated. In CSI, the correlation structure is more complex and governed by wireless physics. A random masking strategy fails to exploit the known structure along the antenna and subcarrier dimensions. Our method explicitly leverages this domain knowledge to create a more challenging and meaningful pre-training task.

- **Beyond NLP and Time Series Models**: While models for natural language (e.g., GPT) or time series may handle 1D sequences, they are not equipped to natively handle the intertwined 3D correlations present in CSI. The success of SigMap hinges on its ability to simultaneously learn representations across these three domains through its tailored pre-training objectives.

In conclusion, the design of SigMap is a principled approach to building a foundation model that respects the unique inductive biases of wireless signal data, rather than forcing the data to conform to architectures designed for fundamentally different modalities.

