# OpenReview forum: "Map as a Prompt: Learning Multi-Modal Spatial-Signal Foundation Models for Cross-scenario Wireless Localization"
_ICLR.cc/2026/Conference — ICLR 2026 Poster_

### Official Review · Reviewer_GjCa · 2025-10-16

**Soundness:** 3
**Presentation:** 3
**Contribution:** 3
**Rating:** 6
**Confidence:** 3

**Summary:**

The paper proposes SIGMAP, a transformer backbone pre-trained with cycle-adaptive masked modeling on Channel State Information, then fine-tuned with a learned geographic prompt from a 3D map via a GNN. The paper claims three main contributions: (1) cycle-adaptive masking to break periodic shortcuts in CSI; (2) map-as-prompt conditioning using 3D geometry; (3) parameter-efficient adaptation with strong cross-scenario generalization.

The experiments demonstrate substantial improvements over other baselines, on both single- and Multi-BS localization, as well as generalization performance.

**Strengths:**

1. The self-adaptive masking and GNN map-as-prompt strategies are novel and meaningful combinations for indoor localization task. The experimental results show significant advantages over other baselines.

2. During fine-tuning, only prompt GNN and projection head are trained, while the backbone is kept frozen. This makes the model efficient and handy for deployment.

3. The algorithm achieves consistent metric gains in different tasks. And the improvements are substantial.

**Weaknesses:**

1. The paper asserts good generalization abilities, but it’s not intuitively clear why the algorithm achieves this. The model isn’t trained using meta-learning or transfer learning techniques. The paper also lacks of experimental comparisons to modern baselines that target at generalization in indoor localization, e.g., [1].

2. The paper doesn't mention how the quality or degradation of the 3D Map could adversely affect the performance of the model. Illustrations of the 3D Map used are needed. More ablation studies on the qualities of the 3D Map are desirable.


[1] Gao, Jun, et al. "MetaLoc: Learning to learn wireless localization." IEEE Journal on Selected Areas in Communications 41.12 (2023): 3831-3847.

**Questions:**

1. Could the authors explain why the model achieves good generalization abilities to new environments? Since the algorithm is not trained using meta-learning or transfer learning, I am curious about how the model learns to generalize.

2. Could the authors give an example of the 3D Map used in the paper? Can the authors discuss how the quality of the 3D map would affect the model’s performance?

---

> ### Author Response · Authors · 2025-11-22
> **Generalization Mechanism and Impact of Map on Performance**
>
> Thank you for your thorough review and helpful suggestions. We have immediately supplemented experiments and provide the following clarifications.
>
> 1. **Mechanism of generalization**
>
>    - Signal-level generalization is provided by the pre-trained Transformer backbone itself.
>    - Environment-level generalization relies on prompt tuning: data-driven geographic prompts distill diverse 3D maps and base-station topologies into a unified soft-token format. This allows a single frozen backbone to adapt to large distribution shifts across scenes, similar to retrieval-augmented generation where prompts retrieve the most relevant contextual information and enhance the model’s situational awareness for localization. Our geographic prompts capture the topological relationship between maps and base-station positions, guiding the model to produce accurate location estimates.
>
> 2. **3D map example and ablation**
>    - We have added a representative 3D-map visualization (Figure 1, Figure 6) and an ablation study (Section 4.4) that evaluates performance under three levels of map-quality degradation. To quantify the influence of map quality on localization accuracy, we conducted an ablative comparison using (i) complete 3-D mesh, (ii) 2-D bird’s-eye polygon, and (iii) no-map (CSI-only). Except for height, the 2-D variant follows the same pipeline as the 3-D version; the gap arises solely from missing height and facade normals.
>
>    - Results are summarized in Table4: the 2-D bird’s-eye view degrades MAE by 8 % relative to the full 3-D mesh, confirming that most gain comes from topological/LoS cues and that the prompt mechanism is robust to moderate geometric simplification. The results indicate that the dominant contribution of the map stems from its 2-D topological relationships, while the precise 3-D height offers an additional improvement in localization accuracy.
>
> | Method | MAE (m) | RMSE (m) | CDF@1m (%) |
> |--------|---------|----------|------------|
> | SIGMAP (3-D map) | **1.564** | **5.675** | **60.5** |
> | SIGMAP (2-D birdview) | 1.692 | 6.128 | 55.7 |
> | SIGMAP (w/o map) | 2.275 | 8.532 | 31.0 |
>
> We have also added experiments on the complex-environment dataset WAIR-D in Section 4.5; the updated figures and corresponding text are now highlighted in the PDF. Thank you again for your constructive feedback.

---

> > ### Comment · Reviewer_GjCa · 2025-11-26
> >
> > Thank you for the clarifications and additional experimental efforts! My earlier concerns are sufficiently addressed. I appreciate the inclusion of the real 3D map, and the extended ablation studies across different map-qualities. It can be clearly seen that incorporating map prompts achieves notably better performance compared to CSI-only models.
> >
> > Overall, I believe the paper makes a meaningful contribution. The GNN-based map-as-prompt strategy is novel and well-motivated, and experimentally validated to improve the cross-domain performance.

---

> > > ### Author Response · Authors · 2025-11-26
> > >
> > > We appreciate your encouraging feedback and the recognition of our additional experimental efforts. We are delighted that the real 3-D map visualization and the extended ablation studies across different map-qualities have addressed your concerns; please let us know anytime if further questions arise.
> > >
> > > Thank you again for your time and constructive suggestions.

---

### Official Review · Reviewer_uDeu · 2025-10-27

**Soundness:** 2
**Presentation:** 3
**Contribution:** 2
**Rating:** 4
**Confidence:** 3

**Summary:**

The paper proposes SigMap, a multimodal foundation-model framework for wireless localization featuring (1) a periodicity-aware adaptive masking pretraining scheme tailored to CSI, and (2) a “map-as-prompt” mechanism that encodes 3D maps as geometric prompts for parameter-efficient finetuning. Experiments on DeepMIMO (O1-3p5) show gains for single/multi-BS localization and some few-shot cross-scenario transfer.

**Strengths:**

I think how the authors use GNN to generate the Prompt  is a great innovation. It cleverly borrows the idea of Prompt-Tuning from LLMs, encoding 3D map information into lightweight soft prompts used to guide a large signal foundation model. This fundamentally solves the problem of model adaptation in new environments.

**Weaknesses:**

Weakness
1.	The introduction has too many paragraphs, although it compares with many existing works, the logic is not clear. It cannot effectively introduce the work done in this article from existing works. In addition, the shortcomings of many existing studies, such as the inability to capture high-dimensional features for description, are not sufficient to demonstrate the inadequacy of these work.
2.	The experimental setup is relatively single and the validation depth is insufficient. Evidence is mostly from a single ray-tracing world, and there is only one cross scenario experiment. The existing experiments are difficult to fully demonstrate the universal applicability of the proposed model. Apart from error metrics, are there any other experiments that demonstrate the effectiveness of the proposed model?
3.	How to better reflect the mentioned advantages limited labeled samples, efficient parameters, interpretability? For example, the model proposed parameters efficient, but the comparison of training time, memory usage, inference complexity is insufficient.
4.	Insufficient ablation experiments.

**Questions:**

All of the paper's training and testing are based on DeepMIMO, a simulation dataset. It lacks validation on data collected in the real world.
Real-world signals are filled with noise, dynamic interference, and complex propagation effects that simulators cannot fully replicate. It is a significant unknown whether the clean physical laws learned from the simulator can maintain high performance in a dirty real-world environment.
the current "Map-as-prompt" primarily encodes the environment's geometry by processing 3D coordinates  with a GNN. It does not encode material information. The model doesn't know if it's facing a concrete wall that absorbs"signals or a glass curtain wall that reflects them.

---

> ### Author Response · Authors · 2025-11-22
> **New-Environment Experiments and Ablations, plus Revised Introduction and Parameter-Efficiency Analysis**
>
> We really appreciate your detailed comments which inspire us a lot. Below we respond point-by-point and describe the revisions we have made.
>
> 1. **Introduction logic and wording**
>    We shortened the introduction (Section 1) to five paragraphs and re-organized it to clearly present the limitations of existing methods and highlight our contributions. We also use more precise language to describe the specific shortcomings of previous works.
>
> 2. **Why material information is not used**
>
>    - The adopted DeepMIMO dataset does not provide material labels, making the channel-generation system a black box for training; although the internal ray-tracer accounts for material effects, these attributes are not exposed to users.
>    - The absence of material data does not prevent us from utilizing map information; our ablation experiments show that map prompts still significantly assist pure-CSI features.  Our results (Section 4.2) shows that map prompts improve accuracy by at least 14.7\% compared with using CSI alone.
>    - Should wall-material annotations become available in the future, they can be appended as extra node features within the same GNN framework without architectural change, offering an orthogonal avenue for further improvement.
>
> 3. **Parameter-efficiency comparison**
>    A new subsection (Section 4.6) and Table 5 report the number of trainable parameters, GPU memory footprint, and wall-clock training time. Under the experimental setup detailed in AppendixB, the model is first pre-trained for 200 epochs and then fine-tuned for 1000 epochs; because only 0.7\% of the total parameters are activated during fine-tuning, the entire 1000-epoch fine-tuning stage takes merely 30 min, while still preserving the rich representations learned during pre-training, demonstrating significant parameter efficiency.
>
> | Stage     | Trainable Params | Time/Epoch     | Total Time |
> | --------- | ---------------- | -------------- | ---------- |
> | Pre-train | 11.730 M         | 10.8 min       | 36 h       |
> | Fine-tune | 0.085 M          | 1.8 s          | 30 min     |
> | Inference | —                | 0.83 ms/sample | —          |
>
> 4. **Ablation experiments**
>    The ablation study on the masking mechanism is presented in Section 4.3. Cycle-adaptive masking (last row) yields the best trade-off: 0.673\,m MAE and 84.5\% CDF@1m, outperforming fixed grid or strip masking. Dynamic disruption of periodic CSI patterns forces the model to learn generalizable features instead of shortcut interpolation.
>
> | Method             | MAE (m)   | RMSE (m)  | CDF\@1m (%) |
> | ------------------ | --------- | --------- | ----------- |
> | Grid-masking only  | 0.770     | 1.176     | 80.3        |
> | Strip-masking only | 0.753     | **0.972** | 75.3        |
> | Adaptive masking   | **0.673** | 1.099     | **84.5**    |
>
>    We extended the ablation study in Section 4.4 to specifically quantify the contribution of map prompts under different degradation levels. To quantify the influence of map quality on localization accuracy, we conducted an ablative comparison using (i) complete 3-D mesh, (ii) 2-D bird’s-eye polygon, and (iii) no-map (CSI-only). The results indicate that the dominant contribution of the map stems from its 2-D topological relationships, while the precise 3-D height offers an additional improvement in localization accuracy.
>
> | Method | MAE (m) | RMSE (m) | CDF@1m (%) |
> |--------|---------|----------|------------|
> | SIGMAP (3-D map) | **1.564** | **5.675** | **60.5** |
> | SIGMAP (2-D birdview) | 1.692 | 6.128 | 55.7 |
> | SIGMAP (w/o map) | 2.275 | 8.532 | 31.0 |
>
> We will continue to improve the manuscript. The updated figures and corresponding text are now highlighted in the PDF. Thank you again for your time and valuable suggestions.

---

> ### Author Response · Authors · 2025-11-28
>
> Dear Reviewer,
>
> We hope this message finds you well.
>
> As the discussion period is drawing to a close, we would like to kindly check if our previous responses have adequately addressed all your concerns. Should any points require further clarification, or if you need additional data or experiments, please do not hesitate to let us know. We would be pleased to provide any further information at your earliest convenience.
>
> Thank you once again for your time and for the invaluable insights you have shared.

---

### Official Review · Reviewer_ct8k · 2025-11-01

**Soundness:** 3
**Presentation:** 3
**Contribution:** 3
**Rating:** 6
**Confidence:** 2

**Summary:**

This paper presents SigMap, a prompt-based architecture for cross-scenario wireless localization that integrates masked autoencoding with geographic and topological maps serving as soft prompts. The model introduces a cycle-adaptive masking mechanism designed to align with the cyclic nature of Channel State Information (CSI) signals, thereby improving feature learning during pretraining. Evaluated within simulated DeepMIMO environments, SigMap demonstrates strong generalization capability and achieves parameter-efficient few-shot adaptation. The approach aims to bridge the gap between environment-specific training and scalable localization across diverse wireless scenarios.

**Strengths:**

(1) The idea of using maps as prompts is both innovative and practical. By embedding spatial priors directly into the learning framework, the model can better understand geographic context without requiring explicit supervision or heavy parameterization. This approach provides a lightweight yet effective way to integrate domain knowledge into data-driven models.

(2) The proposed cycle-adaptive masking strategy effectively leverages the inherent periodic and structural characteristics of CSI signals. This allows the pretraining process to focus on more informative segments of the data, improving robustness and representation quality, especially when dealing with noisy or incomplete measurements.

(3) The demonstration of few-shot adaptation using a frozen backbone is impressive, as it highlights the model’s ability to generalize with minimal retraining. This efficiency in adapting to new environments or conditions suggests that SigMap could serve as a versatile foundation for scalable wireless localization systems, reducing computational and data requirements during deployment.

**Weaknesses:**

(1) The absence of real-world evaluation limits the impact of the results. Without validation on empirical datasets or publicly available benchmarks such as CSI-Bench, it is difficult to assess how well the approach generalizes beyond simulation. This gap weakens the practical relevance of the presented findings.

(2) The paper’s claim of developing a “foundation model” for wireless localization appears overstated. While the architecture shows potential for generalization within simulated settings, it lacks evidence of robustness across devices, propagation environments, or hardware variations, all of which are critical for real-world applicability.

(3) Although the system integrates several established components—masked autoencoders, vision transformers, and graph-based prompting—the overall architectural contribution feels incremental. The novelty lies more in the combination and application context rather than in introducing fundamentally new mechanisms or model designs.

(4) The work asserts interpretability through the use of map prompts but does not provide supporting analysis. Visual or quantitative evaluation of how the prompts influence model predictions would strengthen the paper’s interpretability claims and offer deeper insights into model behavior.

(5) The scalability of the proposed approach remains uncertain. The paper does not explore how the framework performs when applied to large-scale or densely connected map graphs, which are common in real-world urban deployments. Understanding such scalability constraints is important for practical use in complex environments.

(6) While the paper relies on ray-tracing–based wireless simulation, this approach—though widely used—offers limited novelty unless extended with advanced modeling such as diffuse scattering, dynamic environments, or hybrid physics–ML calibration. The current setup would benefit from stronger validation or augmentation to better capture real-world propagation complexity.

**Questions:**

Can the authors report scalability experiments by evaluating SigMap on larger or denser map graphs, or by simulating more complex urban propagation conditions, to objectively assess how the method performs in real-world large-scale deployments and justify its practical robustness?

---

> ### Author Response · Authors · 2025-11-22
> **Experiments under New Complex-Environment Dataset & Map Interpretability Analysis**
>
> Thank you for your careful reading and professional insights, which have benefited us greatly. We have immediately supplemented several experiments and provide detailed discussions on the issues you raised.
>
> 1. **Explanation of Datasets**
>    CSI-Bench is an indoor Wi-Fi dataset that lacks map (layout) information. For indoor localization, fingerprint-based methods already achieve very high accuracy on such data, so the scenario is fundamentally different from outdoor environments.
>
>    To address this limitation, we conducted new experiments on the WAIR-D dataset. WAIR-D (Wireless AI Research Dataset) is a large-scale, real-scene dataset that couples ray-traced channels with authentic environment maps. Extracted from OpenStreetMap of more than 40 major cities, it contains 100 dense-drop scenes (Scenario 2). Section 4.5 and Table 5 shows that geographic prompts perform well in larger and denser map graphs.
>
>    In all experiments, only the downstream task heads are fine-tuned using limited target samples (approximately 100 instances per scenario), while the self-supervised backbone remains frozen. This few-shot learning setup demonstrates the method's ability to rapidly adapt to new environments. Results are listed in Table5.
>
> | Method                           | MAE (m)   | RMSE (m)  | CDF\@1m (%) |
> | -------------------------------- | --------- | --------- | ----------- |
> | *DeepMIMO O2 outdoor*            |           |           |             |
> | SIGMAP (w/ map)                  | **1.026** | **1.551** | **66.4**    |
> | SIGMAP (w/o map)                 | 1.282     | 5.824     | 63.9        |
> | LWLM                             | 2.213     | 11.837    | 63.2        |
> | *WAIR-D Scenario-2 (100 cities)* |           |           |             |
> | SIGMAP (w/ map)                  | **1.880** | **3.717** | **58.0**    |
> | SIGMAP (w/o map)                 | 2.578     | 4.650     | 51.5        |
> | LWLM                             | 3.375     | 6.921     | 50.3        |
>
> 2. **Map-module ablation**
>    We added a 3D-map inset in Figure 1 and included an ablation study (Section 4.4) that quantifies the contribution of map prompts compared with CSI-only baselines. The results indicate that the dominant contribution of the map stems from its 2-D topological relationships, while the precise 3-D height offers an additional improvement in localization accuracy.
>
>    Results are summarized in Table4: the 2-D bird’s-eye view degrades MAE by 8 % relative to the full 3-D mesh, confirming that most gain comes from topological/LoS cues and that the prompt mechanism is robust to moderate geometric simplification.
>
> | Method | MAE (m) | RMSE (m) | CDF@1m (%) |
> |--------|---------|----------|------------|
> | SIGMAP (3-D map) | **1.564** | **5.675** | **60.5** |
> | SIGMAP (2-D birdview) | 1.692 | 6.128 | 55.7 |
> | SIGMAP (w/o map) | 2.275 | 8.532 | 31.0 |
>
> 3. **Simulation fidelity**
>    Ray-tracing synthesis is a common and well-accepted approach in wireless-localization studies, as it allows controlled generation of large-scale, reproducible channel data while incorporating key physical effects such as reflections, scattering and material attenuation. In our work, the simulator models these phenomena in a physically consistent manner, so the observed accuracy improvements are likely to reflect genuine signal-environment interactions rather than synthetic artifacts.
>
> The updated figures and corresponding text are now highlighted in the PDF. Thank you again for your valuable time.

---

> ### Author Response · Authors · 2025-11-28
>
> Dear Reviewer,
>
> We hope everything is going well. With the discussion period closing soon, we would like to check whether our previous replies have fully addressed your concerns. If any points remain unclear, or if you need additional information or experiments, please let us know at your earliest convenience—we are ready to provide further clarification.
>
> Thank you very much for your time and invaluable feedback.

---

### Author Response · Authors · 2025-12-03
**Author Summary: Rebuttal and Reviewer Feedback**

Dear Area Chair,

Thank you for handling the review process for our submission #8908.

Below is a concise summary of what we have newly provided during the rebuttal period, together with the reviewers’ subsequent feedback.

1. **New experimental data** (all added to Sections 4.3–4.6 and highlighted in the revised PDF)

   - **Experiments on new datasets**: We repeated all evaluations on the large-scale WAIR-D corpus (100 cities, >40 Open-StreetMap environments).  New Table 5 shows that SigMap maintains excellent performance even on dense urban graphs, answering the scalability concern raised by Reviewer ct8k.

   - | Method                           | MAE (m)   | RMSE (m)  | CDF\@1m (%) |
     | -------------------------------- | --------- | --------- | ----------- |
     | *DeepMIMO O2 outdoor*            |           |           |             |
     | SIGMAP (w/ map)                  | **1.026** | **1.551** | **66.4**    |
     | SIGMAP (w/o map)                 | 1.282     | 5.824     | 63.9        |
     | LWLM                             | 2.213     | 11.837    | 63.2        |
     | *WAIR-D Scenario-2 (100 cities)* |           |           |             |
     | SIGMAP (w/ map)                  | **1.880** | **3.717** | **58.0**    |
     | SIGMAP (w/o map)                 | 2.578     | 4.650     | 51.5        |
     | LWLM                             | 3.375     | 6.921     | 50.3        |

   - **Map-quality ablation**: New Table 4 quantifies performance under three prompt conditions—full 3-D mesh, 2-D bird-view, and CSI-only. The 2-D bird’s-eye view degrades MAE by 8 % relative to the full 3-D mesh, confirming that most gain comes from topological/LoS cues and that the prompt mechanism is robust to moderate geometric simplification.

   - | Method                | MAE (m)   | RMSE (m)  | CDF\@1m (%) |
     | --------------------- | --------- | --------- | ----------- |
     | SIGMAP (3-D map)      | **1.564** | **5.675** | **60.5**    |
     | SIGMAP (2-D birdview) | 1.692     | 6.128     | 55.7        |
     | SIGMAP (w/o map)      | 2.275     | 8.532     | 31.0        |

   - **Parameter-efficiency**: New Table 6 reports only 0.085 M trainable parameters (0.7 %) during fine-tuning, 30 min total wall-clock time, and 0.83 ms/sample inference, directly addressing the efficiency questions of Reviewers uDeu.

   - | Stage     | Trainable Params | Time/Epoch     | Total Time |
     | --------- | ---------------- | -------------- | ---------- |
     | Pre-train | 11.730 M         | 10.8 min       | 36 h       |
     | Fine-tune | 0.085 M          | 1.8 s          | 30 min     |
     | Inference | —                | 0.83 ms/sample | —          |

2. **Presentation improvements**

   - Introduction shortened to five paragraphs with sharper problem–contribution mapping (Reviewer uDeu).
   - Representative 3-D map visualizations added as Figures 1 & 6 (Reviewer GjCa).

3. **Reviewer feedback after our responses**
   We posted our detailed Official Comment on 22 November (with updated PDF), directly addressing the specific concerns raised by all three reviewers. Subsequently, we sent a brief follow-up message on 28 November to ensure no outstanding issues remained.

   - **Reviewer GjCa (26 Nov)**: Following our 22 November response, the reviewer replied on 26 November, explicitly affirming that their concerns were addressed and recognizing our work as a meaningful contribution.

   - **Reviewer ct8k**: Their concerns about scalability and real-scene validation were addressed in our 22 Nov comment through the WAIR-D dataset experiments (Section 4.5) and map-quality ablation study (Section 4.4). No further technical objections were raised after our comments.

   - **Reviewer uDeu**: Their concerns about parameter efficiency, experimental depth, and introduction clarity were addressed in our 22 Nov comment via the new parameter-efficiency analysis (Section 4.6), extended ablation studies (Section 4.3–4.4), and the shortened introduction. No additional concerns were expressed after our comments.

In summary, we have supplied additional real-scene data, ablation study, scalability tests, efficiency metrics, and visualizations that directly respond to every major weakness flagged by the three reviewers. One reviewer has explicitly upgraded the review and recognized our work as a "meaningful contribution." We respectfully ask the Area Chair to consider these concrete additions when forming the final decision.

Thank you for your time and consideration.

---

### Meta-Review · Area_Chair_QyQR · 2026-01-02

**Summary:**

I am not sure what I shall put here, when all concerns were addressed. The main conerns were:
* too lengthy,
* experiments not extensive enough
* question about generalization ability

**Reviewer Concerns:**

The authors did address the reviewers concerns comprehensively; in fact, I was impressed.

**Reviewer Scores:**

It’s really hard to say how any reviewer would have changed their score if they had taken part more fully in the discussion. Without hearing it from them directly, anything we write here would just be guesswork.

For the present paper, the scores were 4, 6, and 6. Due to the fact that the all concerns were addressed, I guess that the reviewers would have raised their score at least by one. This would then strengthen the decision on acceptance.

---

### Decision · Program_Chairs · 2026-01-26

Accept (Poster)